# Opening a New Era with Machine Learning in Financial Services? Forecasting Corporate Credit Ratings Based on Annual Financial Statements

**Mustafa Pamuk *** and **Matthias Schumann**

Faculty of Business and Economics, University of Goettingen, 37073 Goettingen, Germany;
mschuma1@uni-goettingen.de
* Correspondence: mustafa.pamuk@uni-goettingen.de

**Abstract:** Corporate credit ratings provide multiple strategic, financial, and managerial benefits for decision-makers. Therefore, it is essential to have accurate and up-to-date ratings to continuously monitor companies' financial situations when making financial credit decisions. Machine learning (ML)-based internal models can be used for the assessment of companies' financial situations using annual statements. Particularly, it is necessary to check whether these ML models achieve better results compared to statistical methods. Due to the multi-class classification problem when forecasting corporate credit ratings, the development, monitoring, and maintenance of ML-based systems are more challenging compared to simple classifications. This problem becomes even more complex due to the required coordination with financial regulators (e.g., OECD, EBA, BaFin, etc.). Furthermore, the ML models must be updated regularly due to the periodic nature of annual statements as a dataset. To address the problem of the limited dataset, multiple sampling strategies and machine learning algorithms can be combined for accurate and up-to-date forecasting of credit ratings. This paper provides various implications for ML-based forecasting of credit ratings and presents an approach for combining sampling strategies and ML techniques. It also provides design recommendations for ML-based services in the finance industry on how to fulfill the existing regulations.

**Keywords:** forecasting corporate credit ratings; multi-class classification; machine learning; annual financial statements

## 1. Introduction

A quantified and accurate assessment of corporate creditworthiness provides multiple benefits to external decision-makers such as investors, creditors, and bankers in terms of their financial, strategic, and operational judgments based on ratings (Kao et al. 2020; Pai et al. 2015; Hwang et al. 2010). In addition, all participants and decision-makers are legal entities that cannot act alone but act through certain persons with clear responsibilities (Peráček and Kaššaj 2023). Moreover, the credit ratings have to be updated by rating agencies to reflect the latest credit status of the companies (Abad and Robles 2014; Xia 2014). In addition, annual financial statements are important for several stakeholders in the finance industry while determining the financial situation and development of an entity to guide the current and future of external activities (Bauer and Agarwal 2014; Daniel et al. 2017). In light of financial fluctuations caused by the COVID-19 pandemic, it has become more important to consistently monitor the financial situation of even the most successful international companies and their business partners (Islam 2020). In addition, it is necessary to improve quality and accuracy while forecasting corporate credit ratings (Korol 2019; Wang et al. 2020). However, companies do not always provide accurate information, which degrades automated credit scoring results and must be carefully considered and sorted out (El Kalak and Hudson 2016; Pamuk et al. 2021). In this context, it is necessary to gather

further recommendations for policy-compliant forecasting, monitoring, and maintenance mechanisms (Kim and Ahn 2012; Kao et al. 2020).

As a result of the financial crisis in 2007–2009, banks were already encouraged to develop internal credit rating models for evaluating loans to businesses and their associated risks, as well as for business partners with credit lines (Ala'raj and Abbod 2016). Compared to traditional credit ratings, ML-based approaches have a strong ability to extract meaning and identify trends from datasets (Marqués et al. 2013). These approaches can facilitate the process of identifying key criteria for forecasting credit ratings in the underlying dataset. Obviously, these ML models must be able to predict the correct risk classes or credit rating from a non-binary list. It is essential to consider that annual financial statements have different qualities depending on the size of the company while illustrating the financial health of companies (Andreeva et al. 2016; Ciampi 2015; Pai et al. 2015; El Kalak and Hudson 2016). Since forecasting accuracy is to be increased, non-traditional and up-to-date data from numerous other sources can be considered additionally (Hurley and Adebayo 2016; Onay and Öztürk 2018). But first of all, it is essential to clearly define appropriate design, development, and maintenance processes to address the complexity due to the multi-class classification problem while forecasting credit ratings (Lessmann et al. 2015; Benbya et al. 2020; Faraj et al. 2018; Huang et al. 2012).

Moreover, this classification problem becomes more complex due to the requirements of the financial supervisory authorities for ML-based financial services (BaFin 2021; Benbya et al. 2020; Faraj et al. 2018; European Commission 2021). The fact that any changes in ML-based services must be agreed upon by supervisory authorities poses a critical coordination effort for financial services companies and entities, especially for services with long-term development and testing periods (BaFin 2021; Faraj et al. 2018; European Commission 2021). This kind of coordination with regulatory authorities is not part of traditional credit scoring, which poses a challenge during the development and deployment of ML-based services. Current requirements, regulations, and expectations of the supervisory authorities (e.g., explainability and traceability of ML) increase the complexity of the maintenance and monitoring of ML-based services (Arya et al. 2019; Arrieta et al. 2020). Consequently, research and practice struggle with multiple questions and conceptual regulations, even though better results are achieved through ML.

The benefits or improvements of ML-based credit ratings over existing statistical methods are necessary to examine before building ML-based financial services. Depending on the used dataset and the relevance of periodic changes and releases, it is necessary to monitor and retrain the ML models to ensure a good forecasting quality of credit ratings. In cases with non-binary classification problems, it is elaborate to define clear structures for the design, development, and monitoring of ML-based services and at the same time meet the requirements of supervisory authorities (Pamuk et al. 2021; BaFin 2021; Deutsche Bundesbank and BaFin 2021).

Different from traditional rating agencies, ML-based services can be adapted and trained more dynamically using partial data on a monthly or quarterly basis to track the latest financial developments (Hurley and Adebayo 2016). Furthermore, existing studies on this topic focus more on the ML techniques themselves and do not propose an appropriate structure for the deployment or maintenance of the underlying process (Fraisse and Laporte 2022; Li and Mei 2020; Rodrigues et al. 2022; Lokanan and Sharma 2022; Guerra et al. 2022). Due to the fact that financial services companies have to fulfill the requirements if they want to offer or use ML-based services, they have to consider steps like evaluation and deployment properly (Benbya et al. 2020; BaFin 2021). Furthermore, well-defined structures are necessary to be able to retrain, test, and validate these models for forecasting credit ratings when new annual statements are available.

Financial ratios required for forecasting credit ratings can subsequently be derived and/or (re-)calculated using these annual statements (Obermann and Waack 2016; de Andrés et al. 2012). Hence, we focus on finding key aspects for the use of ML in forecasting corporate credit ratings and associated procedures to support development, monitoring,

and maintenance structures. From this background, we define the following research question in this paper:

*RQ: How can ML and oversampling techniques be used to analyze annual financial statements for forecasting corporate credit ratings?*

First, we introduce the theoretical foundations of corporate credit rating and give an overview of existing European regulatory principles in the finance industry. Subsequently, the research approach for forecasting credit ratings presented in Section 3 outlines each step of the construction and evaluation of the ML approach. In Section 4, we discuss the results of this study. Implications for research and practice and possible future directions are presented in Section 5.

## 2. Theoretical Foundations

In the following, we first summarize the theoretical background of corporate credit ratings. Further, the existing regulatory principles in Europe are presented in Section 2.2 to ensure a common understanding related to ML usage in the finance industry.

### 2.1. Corporate Credit Ratings

The credit rating of a company indicates how high the risk is that a borrower or counterparty will not meet its obligations according to the agreed terms (BIS 2000). As one of the three largest external rating agencies, Moody's Investors Services (2022) defines credit rating as follows: "*Ratings assigned on Moody's long-term and short-term rating scales are forward-looking opinions of the relative credit risks of financial issued by non-financial corporates, financial institutions, structured finance vehicles, project finance vehicles, and public sector entities.*" The definition presented here distinguishes between short-term and long-term credit ratings and presents them as an assessment rather than a legally binding valuation. Moreover, credit ratings can be issued for different types of debtors, such as companies and countries (Gavalas and Syriopoulos 2014).

Credit rating refers to both the procedure and its result (Gavalas and Syriopoulos 2014; van Gestel and Baesens 2009). Moreover, the creditworthiness of a company is assessed as part of the rating process and is an important part of risk management in financial services (Andriosopoulos et al. 2019; Gavalas and Syriopoulos 2014). There is a distinction between internal and external credit ratings based on the issuer of the credit rating (Dimler et al. 2018). Internal ratings are issued by banks as part of the lending process to ensure that equity capital is adequately backed by risk (White 2013; Pai et al. 2015). External ratings are usually performed by external credit agencies or information providers and increase market transparency by preventing information asymmetries between issuers and potential investors (Dimler et al. 2018; Dittrich 2007; Benlala 2023).

Financial ratios are essential in determining the creditworthiness of corporations because they rely on corporate governance characteristics and macroeconomic variables, which provide additional relevant information for ratings (Matthies 2013). Hereby, linear regression, logistic regression, or discriminate analysis are common statistical techniques for forecasting credit ratings (Matthies 2013; Altman and Saunders 1997; Hirk et al. 2022). However, recent developments in ML and increased computing capacity provide a relevant basis to build and test new internal models using learning or ensemble techniques such as Neural Networks (NN) or XGBoost (Chen and Chen 2022; Hirk et al. 2022). However, further research is necessary to test whether these new ML-based forecasting models can deliver more accurate and compliant results or cause more work and costs in consideration of further design, development, monitoring, and maintenance steps.

Moreover, as a result of Basel II and III guidance, banks that meet certain minimum disclosure and supervisory requirements are allowed to use their own assessments of various risk exposures using internal risk rating systems, known as the "internal ratings-based" (IRB) approach (BIS 2005; BIS 2011). Further, it is advantageous and important for financial companies to be able to classify credit ratings independently of rating agencies (Kim and Ahn 2012; Benlala 2023). Nevertheless, qualitative and quantitative elements

are being considered while assigning credit risk ratings to facilitate the assessment of borrowers' creditworthiness over a period of time (van Gestel and Baesens 2009; Saunders and Allen 2010; Gordy 2003; Andersson and Vanini 2008). For this assessment, a transition matrix helps to identify the financial stress and illustrates movements as a center of modern risk management over the years that can be assisted by ML-based models to increase the precision of financial decisions (Altman and Saunders 1997).

### 2.2. Regulatory Principles for the Use of ML in the Finance Industry

In Europe, financial institutions are subject to extensive regulations for the use of ML and Big Data (BD) techniques to determine credit ratings (BaFin 2021; Friedrich et al. 2021; Popa Tache 2022). If any intelligent and automated technology is implemented, numerous supervisory regulations and implications arise (European Commission 2021). Compliance with these principles is also significant in terms of ensuring applicable ethical standards to prevent any discrimination through automated decisions. ML models that act as a kind of non-transparent "black box" will not be accepted by banking supervision (Bathaee 2018; Bauer et al. 2021; OECD 2021; EBA 2020). Therefore, considering and establishing the documentation process is necessary to promote the explainability and traceability of ML models and ML-based services (Tomak 2022). Besides, due to the increasing interest in ML-based services, supervisory authorities need to develop their own AI/ML competencies to be able to monitor and regulate ML-based services in a structured way (Doerr et al. 2021).

However, common expectations of supervisory authorities regarding the use of ML exist with traditional statistical models such as transparency, reliability, accountability, fairness, and ethics (Prenio and Yong 2021; Deutsche Bundesbank and BaFin 2021). However, due to the recent developments in ML, it is not yet clear whether and how regulators and supervisors consider specific risks when it comes to offering ML-based services in the finance industry (Prenio and Yong 2021; EBA 2020; Popa Tache 2022). As an example, the German Federal Bank proposes to focus on features of ML that are novel to current regulation and supervisory practices (Deutsche Bundesbank 2020; European Commission 2021). Furthermore, monitoring and maintenance procedures also become more relevant for practitioners and supervisory authorities, increasing their ability to control the quality of the model (Deutsche Bundesbank and BaFin 2021).

From the European perspective, explainability is a key requirement for all ML-based services to ensure fundamental safety and enable people to trust the system (European Commission 2021). However, the progress made through the use of ML, e.g., in credit scoring, compared to classical statistical mechanisms must be recognizable (Deutsche Bundesbank and BaFin 2021). We summarize the identified regulatory principles from the international regulatory and supervisory authorities and recent studies in three groups (see Figure 1): First, there are general principles that consider basic and partially functional aspects such as the potential problems with bias and differentiation (EBA 2020; Deutsche Bundesbank and BaFin 2021; OECD 2021). In this context, the training datasets play a crucial role because if they are biased, the classification will be biased as well (Wall 2018). The principles of the "Development Phase" contain currently common quality features and assessment mechanisms that must be carried out properly by data miners and documented in a structured manner (Cao 2020; Bathaee 2018; Deutsche Bundesbank and BaFin 2021). Principles D2 and D4 determine the set of allowed ML techniques that are explainable enough depending on the task (Doerr et al. 2021; European Commission 2021). Lastly, principles are listed for the "Application Phase," which considers maintenance, monitoring, and control of the system at an abstract level. When ML-based services work with periodic datasets such as annual statements, principles such as A5 must be considered to coordinate with supervisors so that model changes can be made if necessary (OECD 2021).

General
- G1: Responsible Management
- G2: Establishment of Adequate Risk and Outsourcing Management
- G3: Avoid Bias and Discrimination

Development Phase
- D1: Establishment of a data strategy and data governance
- D2: Compliance with data protection rules
- D3: Ensuring correct, robust and reproducible results
- D4: Documentation for internal and external traceability
- D5: Carrying out an appropriate validation process
- D6: Use relevant data for calibration and validation

Application Phase
- A1: Interpretation and utilization of algorithmic results
- A2: Putting the human in the loop as decision maker
- A3: Establishment of intensive approval and feedback processes
- A4: Establishment of emergency measures
- A5: Ongoing validation, higher-level evaluation and appropriate adaptation of the application

**Figure 1.** Identified regulatory principles for AI-Use in the finance industry.

## 3. Research Approach for Forecasting Credit Ratings

To answer the research question, we employ below a standardized data mining project according to CRISP-DM (Shearer 2000). First, we present the dataset in Section 3.1. Then, we identify the key features for forecasting credit ratings and describe the relevant steps for pre-processing in Section 3.2. Perform oversampling to cope with the class imbalance problem as presented in Section 3.3. Modeling; The evaluation results of the ML-based forecasting are described in Sections 3.4 and 3.5.

### 3.1. Dataset of Annual Financial Statements

For this paper, we have selected the dataset from the study of Pamuk et al. (2021), which consists of 3.3 Mio. entries of annual financial statements from 2000–2012 with 74 metrics. Moreover, this dataset provides evidence that small- and medium-sized companies provide less and mostly unprecise information about their financial situations as part of their disclosures and poses a real practical situation (Pamuk et al. 2021; El Kalak and Hudson 2016; Ciampi 2015).

Following the German Commercial Code §267 and §267a, the dataset is divided into four categories based on the business sizes of the companies in the dataset: Micro-, small-, medium-, and large-sized businesses. The analysis of business sizes before data pre-processing revealed that 95% of the entire dataset belongs to micro- and small-sized businesses. Therefore, this analysis, together with the arguments of Ciampi (2015) and El Kalak and Hudson (2016), suggests splitting the dataset into smaller pieces. Due to the insufficient number of entries after data cleaning, large businesses are excluded from this study.

### 3.2. Data Analysis and Pre-Processing of Annual Financial Statements

The dataset contains the rating class, which ranges from AAA to D, as a dependent feature for ML models. Furthermore, we observed an imbalanced distribution for the rating classes in the dataset (Appendix B). Otherwise, they would not be suitable as explanatory features. In addition, the second step is to examine the missing or incorrect values because they can lead to errors and misclassifications in ML models. Because of the requirement by the German Commercial Code (HGB) to provide different disclosures in their financial statements, we checked whether certain disclosures were systematically missing. This exists, for example, if there is no value for earnings before interest and taxes (EBIT) and

operational results together. However, there are no systematically missing values in the entire dataset.

Afterward, 48 metrics with more than 30% missing, negative, or invalid values were deleted. This is because methods such as imputation with the median or arithmetic mean would have distorted the data. The 13 features containing irrelevant information (e.g., addresses and IDs) were also deleted. In total, 61 of the initial 74 metrics were removed from the dataset. The remaining 13 metrics were examined using 25%, 50%, and 75% quartiles for each rating class to check the relationship between explanatory features and the rating class and whether they are useful for the classification (see Appendix A). In total, six metrics showed a positive or negative trend as the rating level worsened: *Asset Coverage Ratio (ACR), Equity Ratio (ER), Short-term Dept Ratio (STDR), 2nd Degree Liquidity (L2), Liquidity Ratio (LR), and Working Capital (WC)*. Due to the identified Pearson Correlation between LR and L2, we only considered LR as a feature in the dataset (Kim and Kang 2012).

Furthermore, it was not possible to reconstruct the calculation of ACR and WC based on the underlying dataset, so we decided to recalculate them using the following basic metrics: *Net Income (NI), Assets (A), Intensity of Investment (II), and Working Capital Intensity (WCI)*. Therefore, all entries with missing or invalid values in NI, A, II, WCI, L2, ER, and STDR were excluded. Furthermore, four additional features (*Working Capital Ratio, Return on Total Assets, Return on Equity, and Asset Coverage Ratio*) were calculated using NI, A, II, and WCI for forecasting corporate credit ratings (see Table 1) that are relevant for examining a company's capital structure, profitability, and liquidity (Fridson and Alvarez 2022; Júdice and Zhu 2021; Beckman et al. 2007; Pamuk et al. 2021). As a result of the data pre-processing, we have 2.5 M. entries remaining in the cleaned dataset (Table 2) that have an uneven distribution of rating classes (Appendix B).

**Table 1.** Key features for forecasting corporate credit ratings.

| Types | Features | |
|---|---|---|
| Capital Structure | ER | (1) Equity Ratio |
| | STDR | (2) Short-term Debt Ratio |
| | WCR | (3) Working Capital Ratio * |
| Profitability | RTA | (4) Return on Total Assets * |
| | ROE | (5) Return on Equity * |
| Liquidity | ACR | (6) Asset Coverage Ratio * |
| | L2 | (7) 2nd Degree Liquidity |

* Calculated using basic metrics.

**Table 2.** Raw and cleaned datasets.

| Business Size (Asset Size) | Raw Dataset | % | After Data Cleaning | % |
|---|---|---|---|---|
| Micro <350k € | 2,087,867 | 63.1 | 1,443,739 | 56.3 |
| Small 350k €–6 Mio. € | 1,077,832 | 32.6 | 1,028,480 | 40.1 |
| Medium 6 Mio. €–20 Mio. € | 97,253 | 2.9 | 93,044 | 3.6 |
| Large >20 Mio. € | 46,055 | 1.4 | - | - |
| Σ | **3,309,007** | | **2,565,263** | |

### 3.3. Oversampling Structure

As derived from the literature, we first split the dataset into three training sets (micro, small, and medium) using the attribute "business size" to cope with the class imbalance problem in the dataset (Batista et al. 2004; Ciampi 2015; El Kalak and Hudson 2016). After data cleaning, we excluded the large companies in this study because the lower

quality caused by increased missing and invalid values in the remaining 4605 entries was insufficient for a multi-class classification problem.

Due to the uneven distribution of rating classes (Appendix B), it is necessary to over-sample the minority classes to avoid any bias. Hereby, we have oriented the oversampling structure according to Pamuk et al. (2021). So, we used the Synthetic Minority Oversampling Technique (SMOTE) and tested/combined it with the cleaning algorithms Tomek Links (SMOTE-TOMEK) and Edited Nearest Neighbor (SMOTE-ENN, see Table 3). SMOTE oversamples minority classes using synthetically generated observations (Chawla et al. 2002). But, to remove the overlap caused by the SMOTE algorithm, Tomek Links and ENN are used in combination with SMOTE (Pamuk et al. 2021).

**Table 3.** Number of observations in each dataset after oversampling.

|  |  | Business Sizes | | |
|---|---|---|---|---|
|  |  | **Micro**<br>**<350k €** | **Small**<br>**350k–6 Mio. €** | **Medium**<br>**6–20 Mio. €** |
|  | **SMOTE** | 3,147,530 | 3,269,140 | 309,090 |
| **Sampling Strategy** | **SMOTE-Tomek** | 2,813,874 | 3,079,375 | 295,014 |
|  | **SMOTE-ENN** | 835,832 | 1,537,544 | 154,576 |

Tomek links are based on the calculation of the distance between two observations of different classes (He and Garcia 2009). It occurs when no observation of the same class with a shorter distance exists in the dataset for these two observations. Identified Tomek Links are then deleted from the dataset. Each combination, SMOTE-TOMEK and SMOTE-ENN, is used to make clean observations of the majority class. For SMOTE, the default value of k = 5 nearest neighbors was considered while deleting observations of the majority class. The number of observations after oversampling is shown in Table 3 and in Appendix B.

*3.4. Modeling*

ML models that have achieved the best results so far have been identified from the literature for credit rating: Neural Network (NN), XGBoost, Logistic Regression (LR), and Decision Tree (DT). Moreover, ten-fold stratified cross-validation was used to train and test the models of LR, DT, and XGBoost because it ensures balancing the class frequencies in the dataset (Olson et al. 2012). For this, each of the three datasets was divided into ten disjoint test and training datasets (Liang et al. 2015). Due to the complexity of NN, we decided to split the dataset into training (60%), testing (20%), and validation (20%) subsets. Moreover, the validation subset is used to optimize the hyperparameters, and the test subset is used to test the predictive power of the NN model. The next step is the standardization of the features in these subsets so that they have a mean of zero and a standard deviation of one. This is essential to prevent the learning process from being dominated by their scale level and to quickly update the weights in algorithms such as NN (Raschka 2014). The following metrics were then used for evaluation of the prediction quality of the ML models: Accuracy, Recall, Precision, and F1.

Moreover, a Grid Search for XGBoost, LR, and DT was performed to find optimal hyperparameters (Bergstra et al. 2011). The hyperparameters used in Grid Search are shown in Table 4. The hyperband algorithm was performed to optimize the number of neurons in the frist layer of the NN (Li et al. 2018). Each iteration in Grid Search eliminates configurations with lower accuracy in the validation dataset until the best hyperparameter is identified. Overfitting is prevented by early stopping for NN if the validation error does not improve after k = 3 epochs (Montavon et al. 2012; Prechelt 2012). Moreover, Root Mean Square Propagation (RMSProp) is employed as an optimizer to improve the convergence speed and stability of the model training process. RMSProp can stabilize the learning process and prevent oscillations in the optimization trajectory (Huang 2020). An overview of the final hyperparameters for NN, XGBoost, LR, and DT is shown in Table 5.

**Table 4.** Grid search parameters for neural networks, XGBoost, logistic regression, and decision trees.

| Neural Network | | XGBoost | | Logistic Regression | | Decision Tree | |
| --- | --- | --- | --- | --- | --- | --- | --- |
| Parameter | Values | Parameter | Values | Parameter | Values | Parameter | Values |
| 1st Layer Number of Neurons | 10–100 in increments of 5 | Learning Rate | 0.1, 0.2, 0.3 | Inverse of Regularization Strength | 1, 2, 3, 4 | Max depth | 2, 3, 5 |
| | | Max Depth | 2, 4, 6 | Max Iteration of Optimization Algorithm | 10, 50, 100, 200 | Max Feature | Sqrt, auto, log |
| Learning Rate | 0.01, 0.001, 0.0001 | Number of Gradient Boosted Trees | 10, 50, 100 | Penalty (Norm) | L2 | Min Sample Leaf | 1, 10, 100, 1000 |
| | | | | Solver (Optimier) | Lbfgs, newton_cg, sag | CCP Alpha (Pruning Parameter) | 0.0, 0.05, 0.1, 0.15 |

**Table 5.** Hyperparameters considered for Neural Network, XGBoost, Logistic Regression, and Decision Tree.

| Neural Network | | XGBoost | | Logistic Regression | | Decision Tree | |
| --- | --- | --- | --- | --- | --- | --- | --- |
| Parameter | Values | Parameter | Values | Parameter | Values | Parameter | Values |
| 1st Layer Number of Neurons | 90 | Base Score | 0.5 | Inverse of Regularization Strength | 2 | CCP Alpha | 0.0 |
| 1st and 2nd Layer Activation Function | Rectified Linear Unit | Learning Rate | 0.3 | Maximum Number of Iterations | 100 | Criterion | Gini |
| 2nd Layer Number of Neurons | 20 | Max Depth | 6 | Penalty | l2 | Max Depth | 5 |
| 3rd Layer Number of Neurons | 10 | Number of Gradient Boosted Trees | 100 | Solver | newton-cg | Max Features | 7 |
| 3rd Layer Activation Function | Softmax | | | | | Min Samples Leaf | 1 |
| Optimizer | RMSProp | Reg_alpha | 0 | Tolerance | 0.0001 | Splitter | best |
| Learning Rate | 0.001 | | | | | | |

### 3.5. Evaluation of the ML-Approach for Forecasting Credit Ratings

The Attribute-based ML-Approach (see Figure 2) has shown that splitting datasets using business sizes and combining them with oversampling techniques has a major positive impact on model performance for forecasting credit ratings (Pamuk et al. 2021). The results of the four ML models presented in Table 6 are the arithmetic mean of the ten models produced by the ten-fold stratified cross-validation. The features are calculated for each rating class, so the weighted average of each feature, ML model, and business size are reported here. The weighted average is used because there may be slight imbalances in the dataset after data sampling with SMOTE-ENN and SMOTE-Tomek.

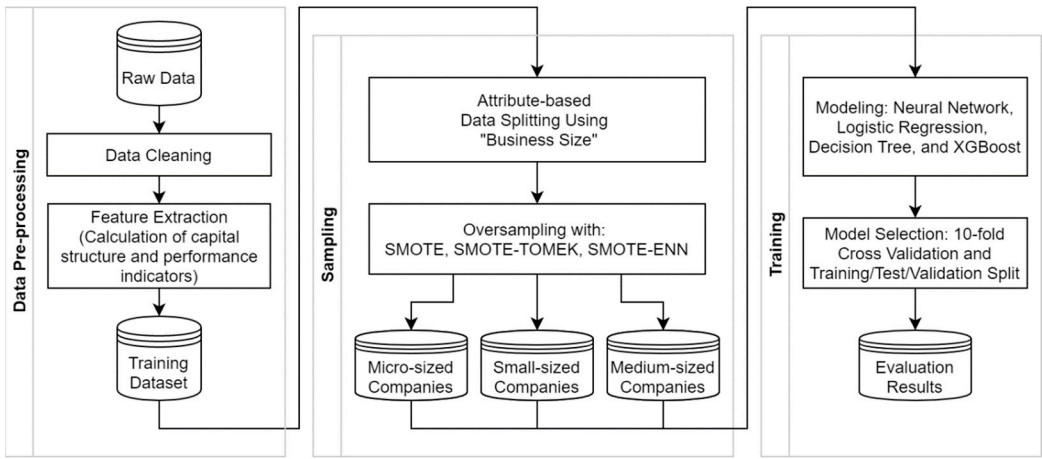

**Figure 2.** Attribute-based machine learning approach.

**Table 6.** Evaluation results of ML models combined with oversampling techniques.

|  |  | NN | | | XGBoost | | | LR | | | DT | | | |
|---|---|---|---|---|---|---|---|---|---|---|---|---|---|---|
|  | **Business Sizes** | **Mic** | **Sm** | **Med** | **Mic** | **Sm** | **Med** | **Mic** | **Sm** | **Med** | **Mic** | **Sm** | **Med** | **Metrics** |
| | **SMOTE** | 0.298 | 0.429 | 0.488 | 0.440 | 0.511 | 0.650 | 0.201 | 0.328 | 0.327 | 0.290 | 0.399 | 0.450 | *Accuracy* |
| | | 0.299 | 0.461 | 0.494 | 0.449 | 0.519 | 0.648 | 0.228 | 0.316 | 0.322 | 0.216 | 0.424 | 0.469 | *Precision* |
| | | 0.298 | 0.429 | 0.488 | 0.440 | 0.511 | 0.650 | 0.201 | 0.328 | 0.327 | 0.290 | 0.399 | 0.450 | *Recall* |
| | | 0.285 | 0.419 | 0.482 | 0.438 | 0.511 | 0.646 | 0.154 | 0.310 | 0.311 | 0.255 | 0.395 | 0.442 | *F1 Score* |
| | **SMOTE-TOMEK** | 0.285 | 0.446 | 0.513 | 0.459 | 0.523 | 0.672 | 0.164 | 0.327 | 0.311 | 0.304 | 0.408 | 0.464 | *Accuracy* |
| | | 0.262 | 0.450 | 0.527 | 0.468 | 0.531 | 0.669 | 0.163 | 0.325 | 0.323 | 0.306 | 0.433 | 0.482 | *Precision* |
| | | 0.285 | 0.446 | 0.513 | 0.459 | 0.523 | 0.672 | 0.164 | 0.327 | 0.311 | 0.304 | 0.408 | 0.464 | *Recall* |
| **Sampling Strategies** | | 0.244 | 0.446 | 0.511 | 0.457 | 0.524 | 0.668 | 0.119 | 0.316 | 0.292 | 0.275 | 0.400 | 0.456 | *F1 Score* |
| | **SMOTE-ENN** | 0.513 | 0.592 | 0.644 | **0.748** | **0.708** | 0.894 | 0.343 | 0.425 | 0.419 | 0.515 | 0.516 | 0.615 | *Accuracy* |
| | | 0.486 | 0.600 | 0.649 | **0.748** | **0.710** | 0.893 | 0.232 | 0.419 | 0.392 | 0.427 | 0.527 | 0.580 | *Precision* |
| | | 0.513 | 0.592 | 0.644 | **0.748** | **0.708** | 0.894 | 0.343 | 0.425 | 0.419 | 0.515 | 0.516 | 0.615 | *Recall* |
| | | 0.475 | 0.589 | 0.634 | **0.747** | **0.707** | 0.893 | 0.210 | 0.388 | 0.385 | 0.460 | 0.510 | 0.594 | *F1 Score* |
| | **SMOTE-ENN (with PR)** | 0.520 | 0.574 | 0.637 | 0.739 | 0.699 | **0.897** | 0.284 | 0.424 | 0.453 | 0.498 | 0.501 | 0.589 | *Accuracy* |
| | | 0.486 | 0.585 | 0.634 | 0.739 | 0.702 | **0.896** | 0.231 | 0.417 | 0.441 | 0.407 | 0.508 | 0.568 | *Precision* |
| | | 0.520 | 0.574 | 0.637 | 0.739 | 0.699 | **0.897** | 0.284 | 0.424 | 0.453 | 0.498 | 0.501 | 0.589 | *Recall* |
| | | 0.469 | 0.571 | 0.629 | 0.737 | 0.699 | **0.896** | 0.194 | 0.395 | 0.417 | 0.443 | 0.493 | 0.560 | *F1 Score* |

Mic: Micro-sized; Sm: Small-sized; Med: Medium-sized businesses. The best results are highlighted in bold.

The models considered in Table 5 were trained, validated, and tested for each of the three oversampling techniques. First, SMOTE was tested to evaluate the forecasting quality. For all four models, it is clear that the forecasting quality is worse than a binary classification problem (Pamuk et al. 2021). This problem arises from the large number of different classes and their associated complexity. Moreover, a comparison with SMOTE and Tomek Links shows that these adjustments can improve the overall prediction accuracy for three out of four models. But for the LR model, the adjustment with SMOTE-Tomek decreases the forecasting quality for each dataset.

Furthermore, the models achieve significantly better results based on the SMOTE-ENN (see Table 6). For each model, the accuracy increases by more than 20%. The number of observations after oversampling for each technique is reported in Table 3 to capture how many of the observations are deleted from ENN as part of the data-cleaning process. It is clear that SMOTE-ENN deleted significantly more entries for each dataset. This could be the reason for better prediction and higher accuracy because deleting the observations makes it easier for ML models to learn the classification boundaries. Moreover, the provisioning rate (PR), as another feature, was calculated and combined with the best-performing oversampling strategy (SMOTE-ENN) to improve the model's performance. But this could only improve the forecasting quality of medium-sized companies with LR and XGBoost models. The reason for this could be the increasing impact of the provisioning

rate depending on the business size (Krüger et al. 2018). For smaller companies, the PR and, consequently, its impact on the financial situation of the company are lower.

An overview of achieved F1-Scores (see Table 7) per rating class helps to analyze whether the best model (XGBoost with SMOTE-ENN) has the same prediction quality after data cleaning. The F1-Scores indicate that the rating classes B-BBB are difficult to classify in the given dataset. In addition, XGBoost has the best F1-Score for rating class D, which is oversampled the most. This could be due to many similar synthetic observations. Moreover, this attribute-based approach (Figure 2) can aid the training and evaluation of multi-class classification cases, such as credit rating, with increased complexity. Hence, it is necessary to ensure enough entries in every minority class (e.g., B-BBB), which poses another issue and encourages further measures while oversampling the dataset. Multi-class classification problems can be extended and handled by one more split using rating classes. Moreover, the results indicate, that the ensemble method XGBoost achieves the best results while forecasting corporate credit ratings.

**Table 7.** F1-Scores for XGBoost with SMOTE-ENN.

| | | Business Sizes | | |
| | | Micro<br><350k € | Small<br>350k–6 Mio. € | Medium<br>6–20 Mio. € |
|---|---|---|---|---|
| | AAA | 0.762 | 0.778 | 0.897 |
| | AA | 0.639 | 0.571 | 0.791 |
| | A | 0.723 | 0.529 | 0.759 |
| | BBB | 0.559 | 0.525 | 0.775 |
| | BB | 0.592 | 0.675 | 0.821 |
| **Rating Classes** | B | 0.634 | 0.681 | 0.859 |
| | CCC | 0.634 | 0.673 | 0.920 |
| | CC | 0.663 | 0.670 | 0.939 |
| | C | 0.797 | 0.730 | 0.953 |
| | D | 0.809 | 0.873 | 0.971 |

## 4. Discussion and Implications for ML-Based Forecasting Credit Ratings

Using an attribute-based ML approach (Figure 2) for forecasting credit ratings indicates that increased complexity due to multi-class classification can be addressed more accurately with oversampling techniques. The pre-processing and splitting of datasets using the attribute "business size" was important to be able to identify and solve the imbalances in the dataset. Moreover, the results confirm that more complex models such as NN and ensemble methods such as XGBoost still provide better forecasting performance (see Table 6). However, the forecast quality of the credit rating depending on business size is lower than a simple binary classification like bankruptcy forecasting (Pamuk et al. 2021). Thus, a sampling based on multiple layers (e.g., based on business size and credit rating class) is necessary in cases like credit rating with multi-class classification. The results illustrate that this approach can iteratively balance datasets based on firm size and additionally based on rating class, which contributes to a more accurate multiclass classification. However, these additional steps for solving the imbalance problem on such periodic datasets will increase the complexity and, thus, the costs of the development and use of ML-based services.

Due to the fact that smaller companies have poorer data quality, a much higher number of observations are removed from the dataset of micro-sized companies (see Table 3) by SMOTE-Tomek and SMOTE-ENN (Ciampi 2015; El Kalak and Hudson 2016). Furthermore, the selection of explanatory features is necessary to provide accurate forecasts for much more complex problems, which will make the development and maintenance of such ML-based services even more challenging. Due to the requirements of regulatory authorities, the underlying methods of ML-based services must also be explainable. Therefore, it is not sufficient to consider only the development of ML models. So, an agreement on underlying procedures, including detailed documentation, with regulatory authorities is mandatory for ML-based services. Even though the dataset has been trained with annual financial

statements, the procedure can be extended or adapted if partial data are submitted, for example, on a monthly or quarterly basis. In this way, it is possible to follow new financial developments more dynamically than with conventional credit ratings.

Moreover, there are regulatory principles (Section 2.2) that must be met to offer an ML-based service at all. The general expectations of the regulatory authorities [G1, G2] must be fulfilled in general, whereby the principle to avoid bias and discrimination [G3] can be tested on a representative dataset before the development phase begins. To build a data strategy and governance [D1], individual steps in the development of ML-based services must be documented carefully [D4] so that reproducibility of the results can be achieved [D3]. Based on the documentation of the steps and measures, the results can be agreed upon with supervisory authorities so that a technical and formal validation [D5] can be carried out in a structured manner. However, whether a relevant dataset is used for validation [D6] is to be verified both by the developers and the regulatory authorities. Therefore, it is relevant to document the intermediate steps and measures and provide the associated datasets.

In addition, this study helps to structure the validation and testing processes of ML models by both developers and regulators and enables iterative identification, discovery, resolution, and documentation of issues [A5]. Structured documentation of each step up to deployment promotes the traceability of the decisions and results [A1], as required by regulatory authorities (OECD 2021; Deutsche Bundesbank and BaFin 2021; Doerr et al. 2021). Considering these regulatory principles and increased efforts, it is obvious to take a critical look at the existing focus on developing ML models in research for different use cases in the finance industry. On the one hand, even if the developed ML models are explainable enough, financial companies must consider and evolve compliant procedures for the development, monitoring, and maintenance of these products or services. On the other hand, they must be able to overcome these challenges for each model change and ensure the consistency and security of their ML-based services. Due to these circumstances, an explainable and well-documented approach must be seen as an important part of ML-based services. The attribute-based ML approach can be used to systematically address the required explainability and promote documentation of the process and decisions for forecasting credit ratings. In this study, we used the built-in feature importance of XGBoost based on 10-fold stratified cross validation to weigh how much the selected features affect the outcome, thus promoting the explainability of the data and the ML model [A1]. It consists of counting the number of times each feature is split across all boosting trees in the model, which can be visualized in a graph with the features ordered according to how many times they appear. In this study, Figure 3 indicates that the features, depending on the underlying sampling technique, features, and dataset, are weighted differently for forecasting credit ratings. It also helps to identify some logical relationships between the features, such as the 2nd-degree liquidity [L2] that plays a more important role while forecasting the credit ratings of smaller companies. Likewise, the equity ratio [ER] indicates a higher importance for small-sized companies but represents the opposite importance level of the short-term debt ratio [STDR] for forecasting credit ratings compared to micro-sized companies.

Explaining a model decision in terms of feature importance can also help to consider the principle [A1] for interpreting the model decisions and providing correct results [D3]. In addition, it is possible to find out whether logical subject-specific relationships can be found that provide for the systematic and rational functioning of the model. From the perspective of developers, it allows us to compare the results of the existing and, if necessary, retrained models with each other [A5]. So, it is possible to examine how the decision-making principles of the models have changed depending on the periodical movements in the dataset, which can bring more transparency into the model's decision-making structure. However, the measures for the required explainability of models vary by ML mechanism and dataset, so it must be considered whether the preferred ML model and measures are sufficiently explainable for regulators.

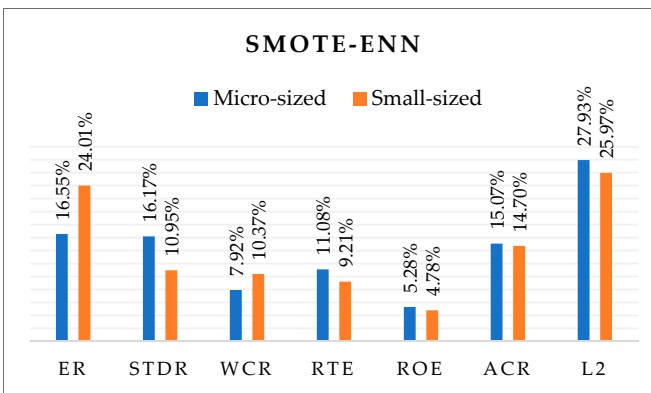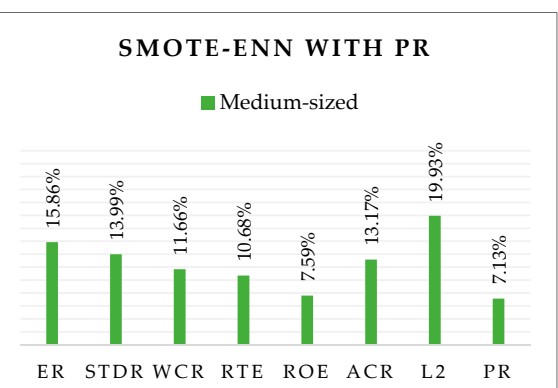

**Figure 3.** XGBoost feature importance overview of the best results.

Furthermore, clearly defined processes are also needed to coordinate with supervisory authorities and enable ML-based services for credit scoring. In this context, providing transparency in the development process can boost mutual understanding of how ML-based services work. In addition, it is important to consider the relevant process by which internal rating models with ML must be developed, tested, agreed upon, and then deployed rapidly to ensure a competitive advantage. Thus, financial services companies and supervisory authorities need to improve their competencies so that this process is completed promptly and adequately (Deutsche Bundesbank and BaFin 2021). The processes of data preparation, model validation, monitoring, and escalation are therefore becoming more relevant than simply focusing on the model's performance in research (Deutsche Bundesbank 2020; EBA 2020). Due to the regulatory principles [A4, A5], ML-based services must be continuously monitored and equipped with emergency measures, e.g., notifying the responsible person in case of outliers in ML-based decisions (Deutsche Bundesbank and BaFin 2021). Such measures are also necessary to control and ensure the quality and accuracy of services provided by financial institutions. Additionally, due to the increasing interest in alternative credit scoring models, non-traditional data from numerous sources and their assessments are also gaining importance (Hurley and Adebayo 2016; Onay and Öztürk 2018). In order to integrate these new data sources into credit rating forecasting, clearly defined processes are essential, especially for transparent and compliant big data analysis (Doerr et al. 2021). Especially, these rapidly evolving market conditions and the associated dynamics require further research and evaluation for the definition of these policy-compliant processes and principles.

From a practical perspective, the split made here based on business size helps to identify the underrepresented and problematic cluster systematically. Thus, the progress made compared to classical statistical mechanisms is recognizable as expected by supervisory authorities (Deutsche Bundesbank and BaFin 2021). Due to the multi-class classification, there is a need for an additional split in the dataset (e.g., rating class) to address imbalances in the dataset more precisely. However, this approach also increases the complexity of developing and maintaining an ML-based system for forecasting credit ratings. Due to the fact that credit ratings are to be forecasted on the most up-to-date dataset, these models must be regularly maintained and monitored. It is also necessary to coordinate with supervisory authorities about whether new models or changes are allowed to be made. Therefore, the concept of a dashboard can extract the results and automatically visualize them to reduce this maintenance effort for financial companies and regulatory authorities. Compared to traditional statistical methods, ML-based approaches can detect patterns and correlations of key features used for forecasting credit ratings between groups, e.g., using business sizes. This allows us to build and use different features depending on the dataset to increase accuracy (Hurley and Adebayo 2016; Marqués et al. 2013).

## 5. Conclusions and Future Work

This paper proposes to find an appropriate structure for ML-based forecasting of corporate credit ratings given the increased complexity that the finance industry, including supervisory authorities, must deal with while developing, deploying, and maintaining ML-based services. Hence, this paper provides multiple implications for research and practice. Within the proposed process, which consists of data pre-processing, sampling, and training processes, we answered the research question. It consists of three oversampling techniques and four ML models for ML-based forecasting of credit ratings. Afterward, we discussed the regulatory principles based on the findings of the research question to highlight the strengths and weaknesses of this approach and identify where future research is needed.

The ML models (NN, XGBoost, LR, and DT) are combined with four sampling techniques to balance the distribution of ten credit rating classes. The results indicate that XGBoost provides the best outcome with SMOTE-ENN (75–89%). However, a closer look at F1 scores after oversampling with SMOTE-ENN per rating class has shown that the rating classes B-BBB are difficult to classify in the given dataset for the XGBoost model. In addition, XGBoost has the best F1 score for rating class D, which is oversampled the most due to a large number of similar synthetic observations. Hence, a split of the dataset based on business size can be extended for multi-class classification problems to increase the forecasting quality. Since new datasets always need to be examined, the proposed approach may be most appropriate for the selected dataset but may differ again for future or other datasets (Marqués et al. 2013). Moreover, this increases the complexity in terms of the performance and maintenance of the developed ML models. To cope with this increased complexity, this study provides an appropriate structure for forecasting credit ratings with an attribute-based ML approach. However, further research is required on how this approach can be extended to consider non-traditional and more up-to-date data for better forecasting credit ratings.

Moreover, the results indicate that ML-based systems working with periodic datasets can be monitored and maintained systematically due to periodic changes. Once the regulatory principles are considered in the development phase of ML-based financial services, only developing further ML models for forecasting credit ratings is not enough for ML-based services in the finance industry. Further research to conceptualize development, documentation, maintenance, and measures to address and fulfill the regulatory principles will shape the future of the financial industry. In addition, traceable and explainable approaches can facilitate coordination with regulatory authorities.

Furthermore, the ensemble XGBoost model could provide better results than the DT, LR, and NN models. However, the increased number of parameters and models, which also increase the training time, must still be viewed critically in terms of the non-ML-based and already-used statistical methods in the finance industry. Nevertheless, the traceability of the processes undertaken by regulatory authorities increases the complexity of development, deployment, and monitoring concepts with ever-changing legal regulations. Compared to traditional credit scoring, there is a high level of dynamism in the development or deployment of new ML-based services. It is essential to bring more structure to the design and development phases of such ML-based services (BaFin 2021; European Commission 2021). This structure can promote more transparency and traceability of ML-based forecasting credit ratings for the supervisory authorities. In addition, the aspect of how to deal with the maintenance of such complex ML-based services is an important issue for future research. But it must be evaluated whether the increased effort for the ML-based services for forecasting credit ratings is profitable in comparison to established procedures.

However, we acknowledge that our research has some limitations and that the identified regulatory principles might not include the principles of all regulatory authorities. We are certain that we have identified the most relevant international authorities and their statements on artificial intelligence, ML, and Big Data. Additionally, this approach to forecasting corporate credit ratings with ML must be critically compared with classical statistical methods in further studies to determine whether the required effort and achieved

benefits indicate more profitable and convincing advantages. Furthermore, it is relevant to investigate further measures based on these regulatory principles so that structured and policy-compliant concepts for the development, deployment, monitoring, and maintenance of ML-based services can be derived.

**Author Contributions:** Conceptualization, M.P. and M.S.; methodology, M.P.; formal analysis, M.P.; data curation, M.P. and M.S.; writing—original draft preparation, M.P.; writing—review and editing, M.P. and M.S.; visualization, M.P.; validation, M.P.; project administration, M.S. All authors have read and agreed to the published version of the manuscript.

**Funding:** We acknowledge support from the Open Access Publication Funds of Göttingen University.

**Data Availability Statement:** The annual financial statement data are not publicly available due to data protection reasons with an external partner.

**Acknowledgments:** We would like to thank the reviewers for their thoughtful comments and efforts towards improving our manuscript.

**Conflicts of Interest:** The authors declare no conflict of interest.

## Appendix A. Financial Ratio Quartiles for the Exploratory Data Analysis and Variable Selection

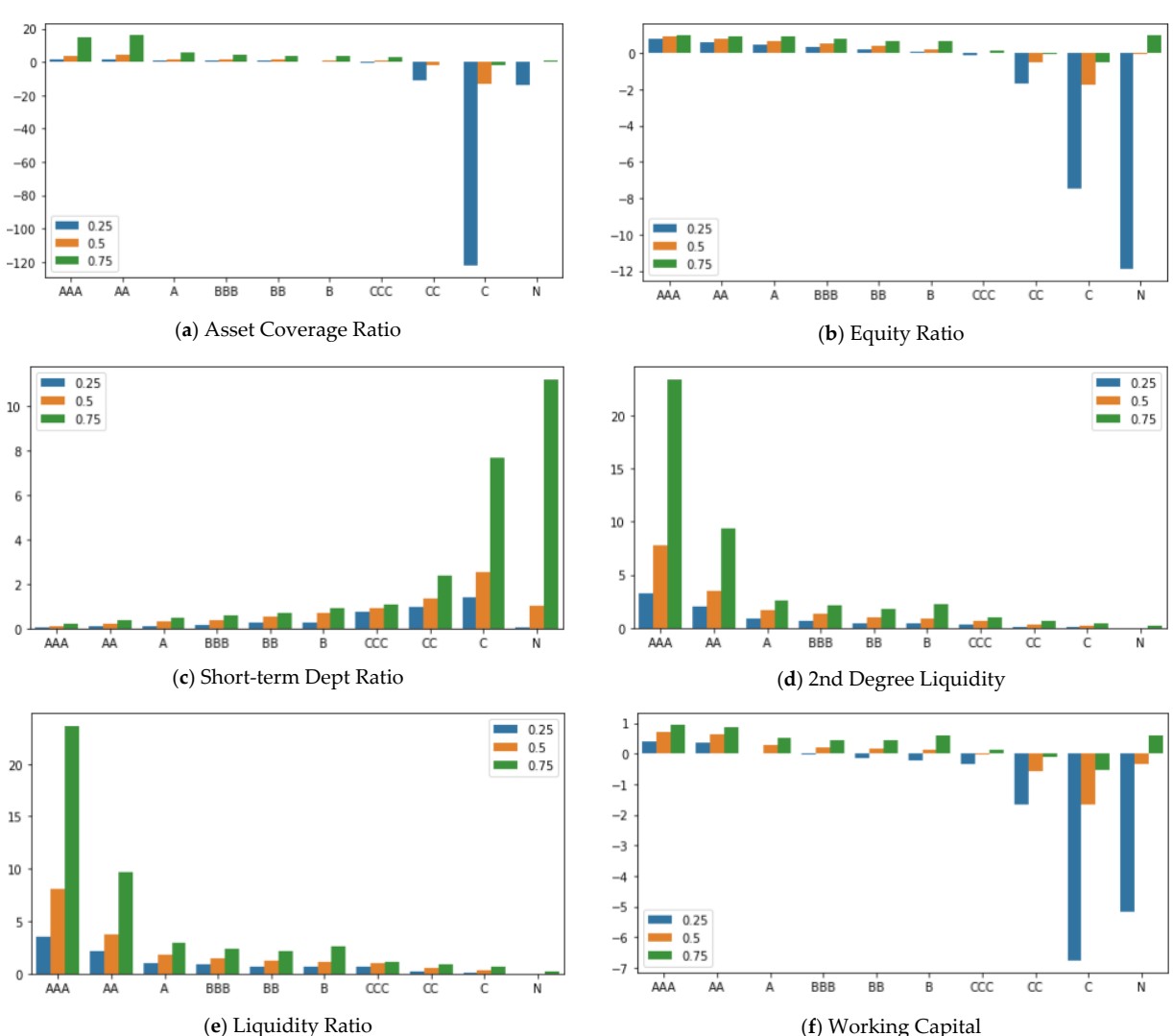

(**a**) Asset Coverage Ratio

(**b**) Equity Ratio

(**c**) Short-term Dept Ratio

(**d**) 2nd Degree Liquidity

(**e**) Liquidity Ratio

(**f**) Working Capital

**Figure A1.** *Cont.*

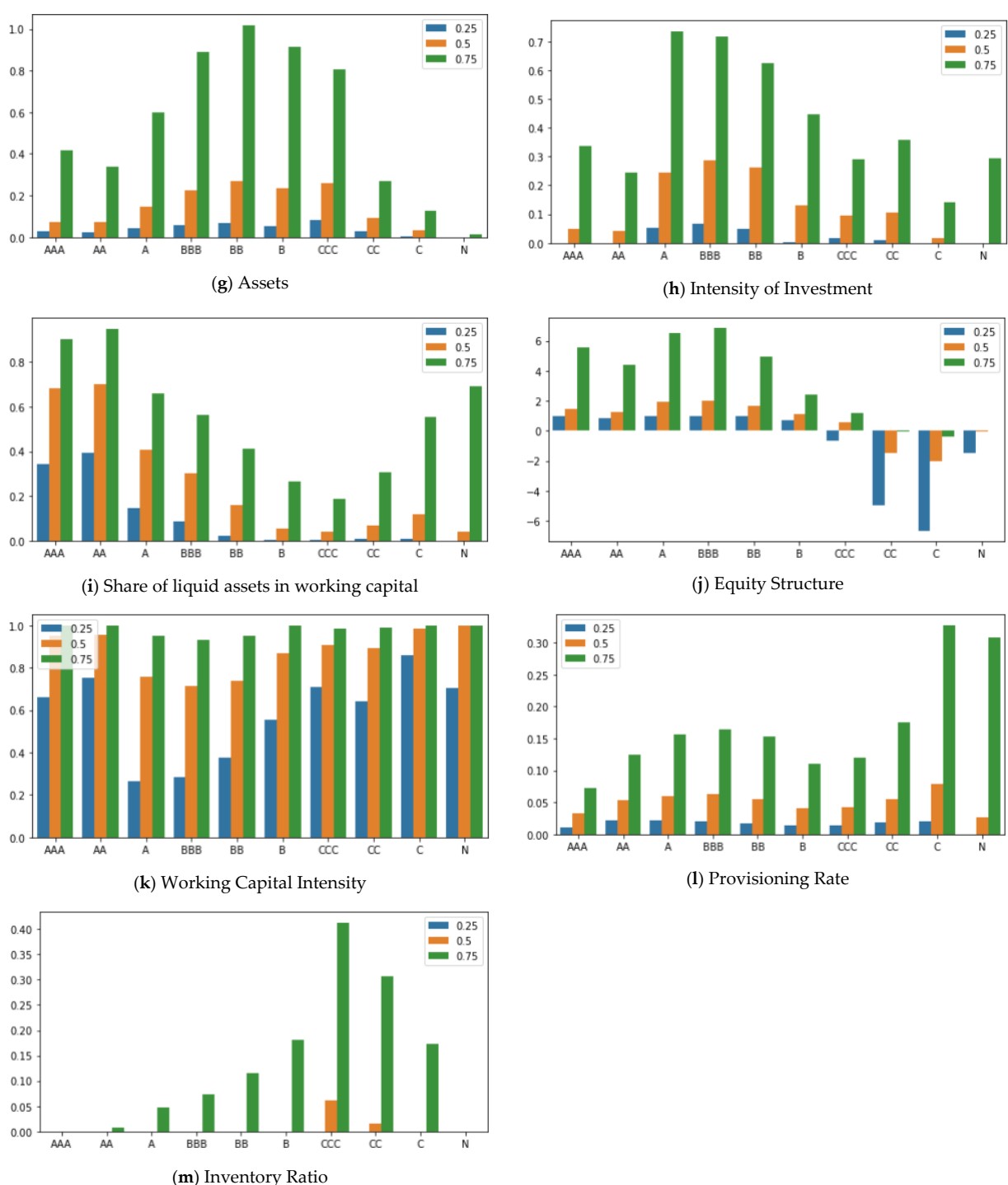

**Figure A1.** Quartiles for exploratory data analysis.

**Appendix B. Sampling Results**

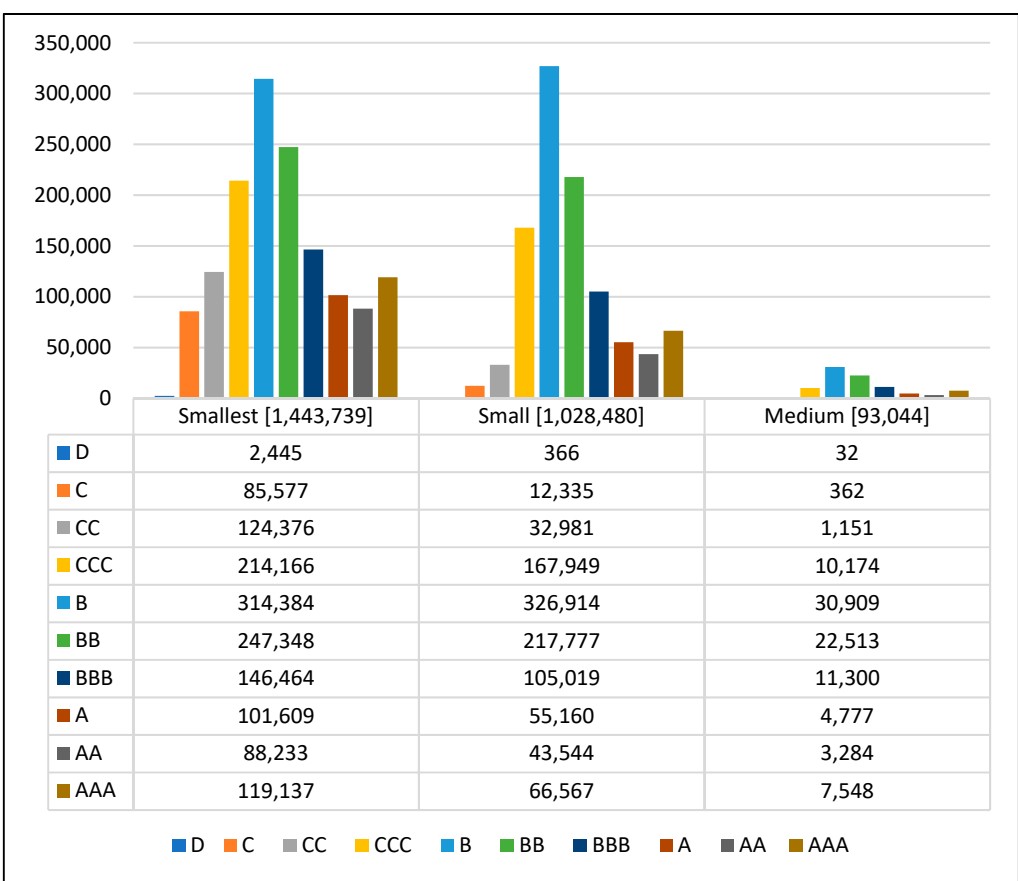

| | Smallest [1,443,739] | Small [1,028,480] | Medium [93,044] |
|---|---|---|---|
| D | 2,445 | 366 | 32 |
| C | 85,577 | 12,335 | 362 |
| CC | 124,376 | 32,981 | 1,151 |
| CCC | 214,166 | 167,949 | 10,174 |
| B | 314,384 | 326,914 | 30,909 |
| BB | 247,348 | 217,777 | 22,513 |
| BBB | 146,464 | 105,019 | 11,300 |
| A | 101,609 | 55,160 | 4,777 |
| AA | 88,233 | 43,544 | 3,284 |
| AAA | 119,137 | 66,567 | 7,548 |

**Figure A2.** Distribution of rating classes in cleaned dataset before sampling.

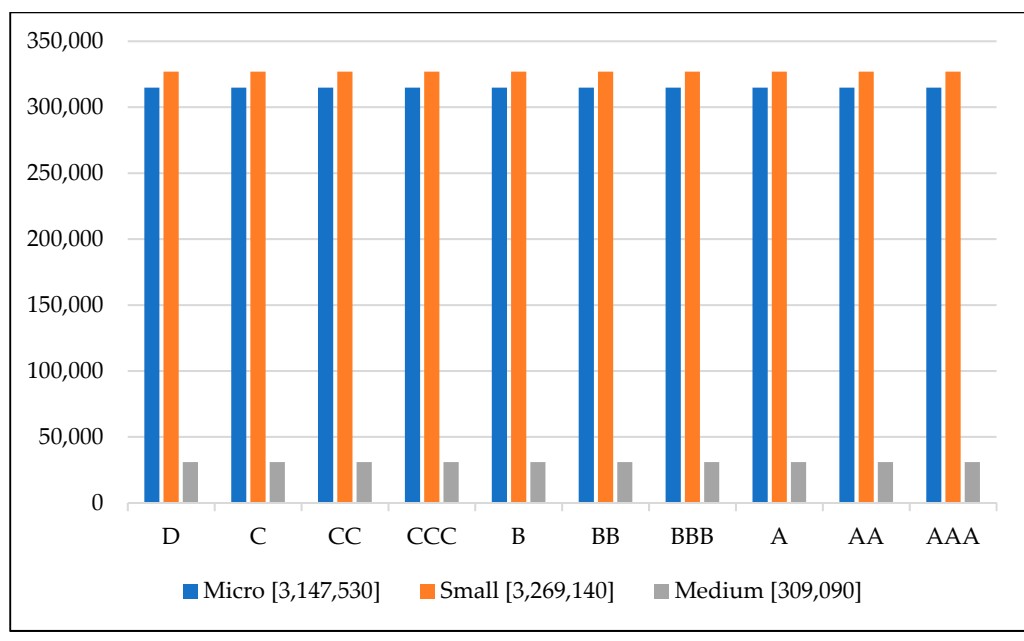

**Figure A3.** Sampling results with SMOTE.

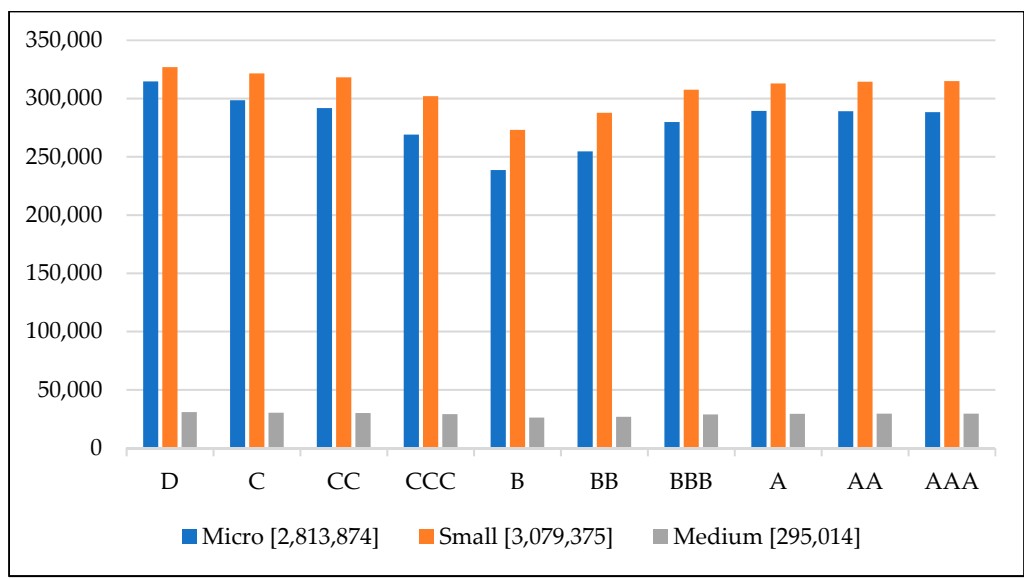

**Figure A4.** Sampling results with SMOTE-Tomek.

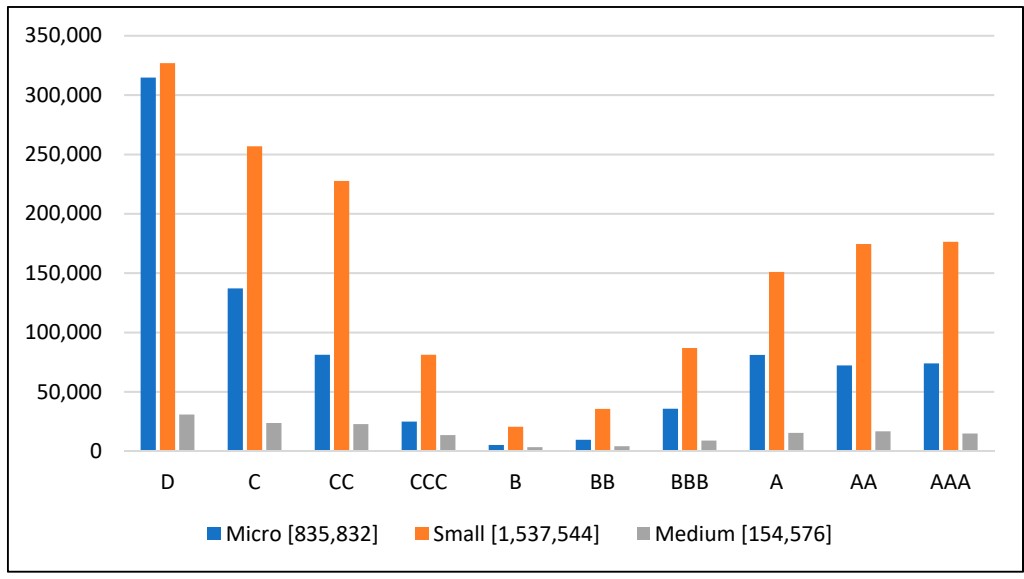

**Figure A5.** Sampling results with SMOTE-ENN.

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
