# Peer review of "Opening a New Era with Machine Learning in Financial Services? Forecasting Corporate Credit Ratings Based on Annual Financial Statements"

_ijfs, doi:10.3390/ijfs11030096_

Round 1
Reviewer 1 Report
The paper offers a comparative ML-based multinomial credit rating forecasting framework to examine the impact of various resampling techniques and predictive modeling approaches on predictive accuracy. The research topic is interesting and actual, as ML based banking modeling is widely discussed among supervisory authorities and market players.
The authors are right to argue that transparency and traceability of ML models are critical. However, in order to avoid that authors fall into their defined trap, the manuscript requires revisions to be transparent and traceable. Please note that a reviewer of a manuscript in this topic acts as a model validator in a supervisory authority.
I have the following recommendations to improve the quality of the paper:
- Distribution of the database per target variable categories should be presented and evaluated. Without this information, it is not convincing whether oversampling was needed at all.
- Variable selection in its form is not transparent (how authors arrived at 7 variables from the initial 74 metrics). How it was determined that financial ratio quartiles showed reasonable behavior according to the target variable? It is natural that some financial indicators have nonlinear and/or non-monothonic relationship with the target variable, so clear positive or negative trend cannot be automatically expected.
- It is OK to discard variables having more than 30% missing values. How were the rest of missing values handled? Imputation?
- Analyze the distribution of target variable categories after oversampling.
- Explain the logic behind partitioning a separate validation and a test subset. Did you use the validation subset to optimize the hyperparameters and the test subset to test the predictive power?
- Setting the first layer in the NN to 90 neurons is excessively high when having only 7 input variables. It may cause worse performance. Explain the applied RMSprop learning method. The abbreviation is not even defined.
- How did you figure out the final prediction from the 100 boosted trees in the XGBoost model?
- Explain the Newton-cg solver algorithm when developing the logistic regression model. How did you decide on this solver from the examined three alternatives? What were the results of parameter and model testings in the logit model?
- Elaborate the applied decision tree method. From the Gini impurity measure I can think of the CART. I have to remark that setting the minimum sample in leaf to 1 is not a valid stopping criterion, and can lead to overfitting.
- Resolving the black box problem is critical in ML modeling. Methodology behind feature importance values on page 12 needs to be in-depth elaborated. I can only have a guess that these figures might be derived from the normalized predictor importance statistics originally published by Saltelli.
Only minor copyediting is needed.
Reviewer 2 Report
I am glad that I can assess this type of scientific study, the content of which is very interesting and current, especially in today's digital age and the impending financial crisis. The 17-page long paper really provides enough space, but (perhaps it's just my subjective opinion) the unedited and broken tables in the text bother me, I recommend the authors to reformat them or at least comb your hair somehow.
The author or authors as non-lawyers are clearly unaware of one essential fact. Banks and other investment trading companies are legal entities and therefore only "constructed social entities" for which their manager must act, i.e. j. natural person - a person with all rights and obligations and, in particular, responsibility for the actions of the legal entity he represents. This fact needs to be explained right at the beginning, while authors such as: Peráček T. & Kaššaj M. (2023) deal very well with this issue. The influence of jurisprudence on the formation of relations between the manager and the limited liability company. Legal Tribune. 13 (1), pp. 43-62. 10.24818/TBJ/2023/13/1.04
In the introduction, the authors also set a research question, but in the end I would appreciate a clear answer to it.
Chapter 2 is really sufficiently elaborated in terms of this type of scientific study, but in order to increase the scientific level, I would supplement it with works such as:
Mejd Aures BENLALA, (2023). Perspectives on Fractional Reserve Banking and Money Creation/Production through the Lenses of Legal and Religious Moral Precepts and Ethics. 2023. Perspectives of Law and Public Administration, 12 (1) pp. 5-30
as well as
Cristina Elena POPA TACHE. (2022). Public International Law and FINTECH Challenges. Perspectives of Law and Public Administration, 11 (2), pp. 218-225
I am convinced that these are relatively undemanding changes that, if the authors implement, this scientific study of theirs will have a higher added value.
Round 2
Reviewer 1 Report
Authors have made progress to improve the quality of their paper. The newly elaborated appendix provides distribution analysis of the database from multiple aspects to visualize the behavior of variables and justify the necessity of oversampling. The revised manuscript adequately clarifies the details of variable selection, the applied data preparation/transformation steps and the characteristics of the dataset as per the target variable.
The revised manuscript better clarifies the selection of hyperparameters for ML model designs, and the modeling strategy behind applying cross-validation and/or partitioning is now clear. Description of modeling algorithms now meets the requirements of the paper. In a scientific journal it is not an acceptable manner to simply argue that Python calculated something using an abbreviated algorithm. It does not matter what programming language, statistical or data mining authors use when developing models, but the assumptions, the essence of operations behind the algorithms should always be 'ruled' on behalf of the authors.
I have a final remark that does not effect the results of the decision tree model: the minimum sample (it can be a minimum number of records or % of the dataset) in a leaf is a very important hyperparameter of any decision tree, as it helps prevent overtraining, not letting further splitting if too little number of observations would get onto the new leaf. Please do not mix better predictive power with avoiding overtraining, since both are important aspects in ML-based modeling. Hence, I recommend not to write in the final version that the stopping criterion was 1, as it is not about increasing classification accuracy but avoiding the specialization to your current dataset.
All in all, authors replied all my comments and the necessary revisions have been made. Therefore, I accept the manuscript and recommend its publication in International Journal of Financial Studies.
Only minor copyediting is needed.
Reviewer 2 Report
It can be seen that the authors have really completely revised the article according to not only my instructions. I agree to the publication of the revised article.